# Current Insights into the Radiobiology of Boron Neutron Capture Therapy and the Potential for Further Improving Biological Effectiveness

**DOI:** 10.3390/cells13242065

**Published:** 2024-12-13

**Authors:** Leah D. Punshon, Maria Rita Fabbrizi, Ben Phoenix, Stuart Green, Jason L. Parsons

**Affiliations:** 1Department of Cancer and Genomic Sciences, College of Medicine and Health, University of Birmingham, Birmingham B15 2TT, UK; ldp570@student.bham.ac.uk (L.D.P.); m.r.fabbrizi@bham.ac.uk (M.R.F.); 2School of Physics and Astronomy, University of Birmingham, Birmingham B15 2TT, UK; b.phoenix@bham.ac.uk; 3University Hospitals Birmingham NHS Foundation Trust, Birmingham B15 2TH, UK; stuart.green@uhb.nhs.uk

**Keywords:** boron neutron capture therapy, DNA damage, DNA repair, ionising radiation, linear energy transfer, radiotherapy

## Abstract

Photon (X-ray) radiotherapy is the most common treatment used in cancer therapy. However, the exposure of normal tissues and organs at risk to ionising radiation often results in a significant incidence of low-grade adverse side effects, whilst high-grade toxicities also occur at concerningly high rates. As an alternative, boron neutron capture therapy (BNCT) aims to create densely ionising helium and lithium ions directly within cancer cells, thus sparing the surrounding normal cells and tissues but also leading to significantly more effective tumour control than X-rays. Although very promising for patients with recurring and highly invasive tumours, BNCT does not currently have widespread use worldwide, in part due to limited and reliable neutron sources for clinical use. Another limitation is devising strategies leading to the selective and optimal accumulation of boron within the cancer cells. Boronophenylalanine (BPA) is currently the major compound used in BNCT which takes advantage of the amino acid transporter LAT1 that is overexpressed in a number of human cancers. Additionally, there is a lack of in-depth knowledge regarding the impact of BNCT on cellular DNA, and the molecular mechanisms that are responsive to the treatment, which are important in developing optimal therapeutic strategies using BNCT, are unclear. In this review, we highlight the current knowledge of the radiobiology of BNCT acquired from in vitro and in vivo studies, particularly in the context of DNA damage and repair, but also present evidence of established and new boron-containing compounds aimed at enhancing the specificity and effectiveness of the treatment.

## 1. Introduction

Boron neutron capture therapy (BNCT) is a cancer treatment that utilises the interaction between stable atoms of boron-10 (^10^B) and thermal energy neutrons (Figure 1) [1,2]. ^10^B is accumulated in cancer cells using boron-containing drugs which are designed to preferentially and specifically target these cells, before irradiating the tumour area with epithermal neutrons that thermalise upon entry into tissues. Thermal energy neutrons are capable of passing through tissues without causing any damage until they are captured by stable nuclei such as hydrogen (^1^H), nitrogen (^14^N) or boron (^10^B). Although ^1^H and ^14^N are present throughout tissues, the probability of neutron capture by these atoms is relatively small. Barns are a measure of the effective cross-sectional area that an atom presents to absorption and is used as a measure of the probability of neutron capture. The capture cross sections of ^1^H and ^14^N are 0.33 barns and 1.7 barns, respectively, whilst ^10^B has a capture cross section of 3990 barns [3]. The interaction between ^10^B and thermal neutrons results in the excitation of stable ^10^B to unstable ^11^B, which ultimately decays, releasing high linear energy transfer (LET) and densely ionising alpha particles (^4^He, ~150 keV/µm) and recoiled lithium ions (^7^Li, ~175 keV/µm) with a tract length of 4–5 µm and 9–10 µm, respectively. LET refers to the amount of energy deposition per section of the radiation track, and the large energy deposition delivered by high-LET radiation has been shown to result in complex/clustered DNA damage (CDD), defined as two or more DNA lesions occurring within one or two helical turns of the DNA [4,5]. This increase in CDD leads to a significantly enhanced relative biological effectiveness (RBE) of BNCT compared to low-LET radiation, such as X-rays and γ-rays, which therefore drives increased tumour cell killing [6].

Another distinct advantage of BNCT over conventional X-ray radiotherapy, apart from increased LET and RBE, is that it can be targeted specifically to tumour cells, limiting damage to the associated normal tissues and organs at risk. This is provided that sufficient ^10^B tumour/normal cell and tumour/blood ratios are maintained. Indeed, boronophenylalanine (BPA), which is currently used in clinical BNCT treatments, is specifically accumulated by tumour cells that overexpress the L-type amino acid transporter, LAT1. This specificity in BPA uptake leads to tumour cell killing and the sparing of normal tissues following thermal neutron irradiation. In contrast, X-rays deposit most of their radiation upon entry into the tissue, with the relative dose lessening as they penetrate deeper into tissues before reaching the tumour. While this is now often mitigated by the modulation of the beam intensity profile and delivery from many directions, it can still be challenging to balance the probability of cure against the risks of unwelcome toxicity for patients. As a result, in some patients, significant damage to the surrounding healthy tissues occurs, leading to significant acute and long-term adverse side effects for patients treated with X-rays [4,7].

Despite there being a general understanding of the radiobiology of BNCT, little is known about the precise effects at the molecular level and how tumour cells respond to the increased complexity of DNA damage induced by BNCT. In this review, we will summarise the major DNA damage repair pathways and proteins employed in human cells, particularly in relation to CDD and BNCT. We also reveal the current understanding of the cellular response to BNCT acquired from preclinical and in vitro studies and discuss the currently utilised boron compounds as well as those actively in development. Understanding more about how the cellular environment is affected and responds to BNCT is important in the development of more targeted boron compounds, as well as potential patient-specific treatment strategies such as combination therapies, which can lead to the optimal utilisation of BNCT in cancer treatment.

## 2. DNA Damage and the Effect of Linear Energy Transfer

Ionising radiation (IR) causes its cell killing effects through creating DNA damage and generating lesions including DNA base damage, DNA single strand breaks (SSBs) and, most importantly, DNA double strand breaks (DSBs) as well as CDD. Following DNA damage induction, cells trigger the cellular DNA damage response (DDR) that involves several DNA repair mechanisms to restore genome integrity (Figure 2). These repair pathways are tightly co-ordinated and regulated by specific proteins and mechanisms.

### 2.1. DNA Base Damage and Single-Strand Break Repair

DNA base damage and SSBs are recognised and repaired by proteins within the base excision repair (BER) pathway. BER is initiated by the recognition of the damage by one of eleven damage-specific DNA glycosylase enzymes which remove the damaged base through cleaving the N-glycosidic bond, leaving behind an apurinic/apyrimidinic (AP) site. AP endonuclease 1 (APE1) is then recruited to the AP site and incises the DNA backbone to generate a SSB with a 5′-deoxyribose phosphate (dRP) and a 3′-hydroxyl end. SSBs generated as intermediates of BER, or through the direct action of IR, are recognised and protected by poly(ADP-ribose) polymerase 1 (PARP-1). PARP-1 also undergoes poly (ADP-ribosyl)ation that creates a scaffold for the recruitment of further repair proteins whilst also allowing PARP-1 to dissociate from the DNA. DNA polymerase β (Pol β) removes the 5′-dRP moiety and subsequently fills the gap in the DNA through the insertion of an undamaged base. The final stage of BER involves the ligation of the remaining nick in the DNA backbone by the XRCC1/DNA ligase IIIα complex [8,9].

### 2.2. DNA Double-Strand Break Repair

DSBs are more toxic to cells than base damage and SSBs, and three repair pathways exist to manage their repair. DSBs are initially recognised by the MRN (MRE11-RAD50-NBS1) complex which is important in the initial processing of the DNA ends, as well as recruiting other repair proteins. MRE11 is an exonuclease required for DNA end processing, generating single stranded DNA (ssDNA) which promotes the recruitment and activation of ATM [10,11], an important kinase in transducing a signalling cascade that leads to cell cycle arrest, DNA damage repair and, in the event of irreparable damage, apoptosis [12]. ATM phosphorylates histone H2AX (γH2AX) which leads to the recruitment of the mediator of DNA damage checkpoint protein 1 (MDC1) which in turn is phosphorylated by ATM [13]. Phosphorylated MDC1 interacts with RNF8 to localise the two proteins to the damage site. RNF8 is an E3 ubiquitin ligase that alongside other E3 ubiquitin ligases promotes the accumulation of DNA damage proteins by ubiquitylating histones [14,15]. Importantly, BRCA1 and 53BP1 are recruited to DSBs in part due to histone ubiquitylation. The ubiquitylation of histones H2A and H2B has been shown to increase the binding of RAP80, which forms a complex with BRCA1, thus increasing BRCA1 foci formation at break sites [16]. BRCA1 and 53BP1 are important proteins in determining which repair pathway will be utilised to resolve DSBs. BRCA1 promotes homologous recombination (HR) repair by excluding 53BP1 in S-phase cells where homologous DNA is available to serve as a template for repair [17]. Alternatively, 53BP1 is a positive regulator of non-homologous end joining (NHEJ) by preventing DNA end resection which is required to initiate HR [18].

#### 2.2.1. Canonical Non-Homologous End Joining (cNHEJ)

Since NHEJ does not rely on a template for repair, it is predominately utilised in the G_1_ phase of the cell cycle and is a relatively error-prone pathway for repairing DSBs which can result in insertions and deletions at the break junction. cNHEJ is initiated by the binding of the Ku70/Ku80 heterodimer to the DNA ends, which prevents DNA end resection and therefore suppresses HR. Ku80/Ku80 also facilitate the recruitment of other repair factors, particularly DNA-dependent protein kinase catalytic subunit (DNA-PKcs). DNA-PKcs caps the free ends of the DNA, then undergoes autophosphorylation and phosphorylates other proteins, leading to the recruitment of processing enzymes and repair factors. These include nucleases such as Artemis which can generate compatible ends for ligation [19,20]. Where a DSB has resulted in blunt ends, minimal processing will likely be required prior to repair. NHEJ can utilise short regions of microhomology (typically ≤ 4 nucleotides) between the two DNA ends to guide the joining of the DNA ends, particularly for breaks with 5′- and/or 3′-DNA overhangs. This microhomology can either be pre-existing or generated during the end-processing phase [20]. Missing nucleotides are synthesised by DNA polymerases lambda (Pol λ), mu (Pol μ) and terminal deoxynucleotidyl transferase (TdT) [21]. Once the DNA ends are sufficiently processed and compatible, the XRCC4-DNA ligase IV complex is recruited to the break, stabilised and activated by XLF to perform DNA ligation.

#### 2.2.2. Alternative Non-Homologous End Joining (aNHEJ)

Cells have been shown to utilise another NHEJ pathway for the repair of DSBs called alternative NHEJ (aNHEJ), also known as microhomology-mediated end joining (MMEJ). aNHEJ may also be utilised under other circumstances, but its full capabilities are yet to be elucidated. aNHEJ functions independently of Ku, DNA-PKcs and DNA ligase IV, instead utilising small regions of microhomology to direct repair. The pathway begins with the binding of PARP-1 to the DNA ends followed by short-range DNA end resection by MRE11 and CtIP. The recruitment of the XRCC1/DNA ligase IIIα complex or DNA ligase I leads to the ligation of the DNA ends. DNA polymerase theta (Pol θ) has also been implicated in aNHEJ by promoting the end joining reaction. This repair pathway acts slowly and, due to its use of microhomology, is error-prone and associated with a high burden of mutations and chromosomal aberrations [22,23].

#### 2.2.3. Homologous Recombination (HR)

Since HR requires the sister chromatid to be used as a template for repair, this pathway is restricted to the S/G_2_ phases of the cell cycle. HR is initially driven by the production of ssDNA at the DSB site which is promoted by the MRN complex alongside the endonuclease CtIP [24]. MRE11 and CtIP resect the DNA ends to produce short 3′ overhangs. Other nucleases such as EXO1 and DNA2 are then recruited to the break site through interaction with the MRN complex and facilitate the long-range bidirectional resection of the DNA ends [25,26]. The ssDNA overhangs produced are rapidly coated with replication protein A (RPA) to protect the DNA ends from degradation and prevent secondary structure formation. ATR interacting protein (ATRIP) can associate with RPA-coated DNA, a process that is essential for the recruitment of the ATR protein kinase to the damage. The kinase activity of ATR is crucial for the activation of downstream signalling pathways that modulate the recruitment of DNA repair proteins, as well as cell cycle checkpoint activation [27]. RPA is displaced from the ssDNA and replaced by RAD51, a process stimulated by BRCA2. This creates nucleoprotein filaments that are essential for the subsequent homology searching and strand invasion steps of HR. BRCA2 also stabilises the nucleoprotein filaments by inhibiting RAD51 ATPase activity, which prevents the disassembly of the filaments [28,29]. The RAD51-ssDNA filament subsequently searches for homologous DNA in the sister chromatid to use as a template for repair, producing a D-loop structure. RAD51 dissociation from duplex DNA is regulated by RAD54 which then enables DNA synthesis to occur, catalysed by DNA polymerases. Second-end capture leads to synthesis from the 5′ ssDNA end, and the ligation of the newly synthesised DNA strands leads to the formation of a double Holliday junction [30]. The dissolution of the Holliday junction by helicases results in a non-crossover product whereby the homologous chromosomes are resolved without exchanging DNA. Alternatively, resolution can result in both crossover and non-crossover products and utilises structure-specific endonucleases that symmetrically cleave the DNA to generate two nicks across the Holliday junction which can be repaired by DNA ligase [31,32].

### 2.3. Complex DNA Damage Repair

CDD is defined as two or more DNA lesions within 1–2 helical turns of the DNA, often caused by a single track of IR. CDD can be broadly divided into non-DSB and DSB-associated, of which the former is considered to predominate [33]. The complexity of the damage produced by IR is strongly linked to the LET, where high-LET IR (such as helium and carbon ions) generates more ionising events and therefore has a greater propensity to generate more CDD than low-LET IR (such as X-rays/γ-rays). The structural and chemical complexity of CDD presents a greater challenge to the cellular DNA repair machinery, and in the case of DSB-associated CDD, is likely to involve multiple proteins and pathways. Therefore, CDD is considered to involve delayed repair and its persistence in DNA, contributing greatly to increased mutation rates as well as reduced survival following IR treatment [34,35,36].

The production of two high-LET particles in the neutron capture reaction, ^4^He and ^7^Li, means that the cellular DNA damage caused following treatment with BNCT is highly complex. Despite this, surprisingly, the specific nature of the DNA damage and the pathways that govern its repair has not been studied in any level of detail. One study investigated DNA damage repair in Huh7 human hepatocellular carcinoma cells, revealing that boric acid-mediated BNCT induced DSBs and that key proteins involved in the HR pathway, BRCA1 and RAD51, were upregulated following treatment, indicating a reliance on HR [37]. Similarly, another study showed an upregulation of *rad51* and *rad54* mRNA following BPA-mediated BNCT compared to untreated controls in both WHO human thyroid cancer cells and Mel J human melanoma cells [38]. Conversely, other studies have shown that BNCT-induced DSBs do rely on NHEJ for repair. It has been shown that xrs5 Ku80-deficient Chinese hamster ovary (CHO) cells showed increased DSBs and sensitivity following boric acid- and BPA-mediated BNCT compared to wild-type CHO-K1 cells [39]. Similarly, it has been demonstrated that DNA ligase IV (Lig4^−/−^p53^−/−^) defective mouse embryonic fibroblast cells exhibited higher sensitivity to BPA-BNCT compared to DNA ligase IV (Lig4^+/+^p53^−/−^) proficient cells [40]. The results of these studies therefore lead to conflicting data as to whether NHEJ or HR may play major roles in the repair of BNCT-induced DNA damage, although there may also be cell line and tumour-specific responses which need to be understood. The utilisation of different neutron sources and radiation dose delivery in the proportion of thermal, epithermal and fast neutrons delivered to cells, as well as differing levels of low-LET γ-radiation, may furthermore complicate the interpretation of these studies. It is therefore important that more systematic studies using well-characterised BNCT facilities and cell models are performed to explore the levels and complexity of the DNA damage generated by BNCT and additionally to understand the cellular DNA repair pathways that mediate the response.

## 3. Clinical Boron Compound Delivery

Clinical studies investigating BNCT began in the 1950s primarily on patients with glioblastoma, which utilised non-tumour-selective drugs such as boric acid and ^10^B-enriched borax [3]. The drugs were delivered to patients via intravenous injection, and neutrons were generated by attenuating the beam of nuclear research reactors to produce thermal neutrons. Due to the non-tumour-specific nature of the boron carriers utilised and the poor tissue penetrance of the thermal energy neutrons, initial studies showed little success. To improve the treatment, primary debulking craniotomies followed by intra-operative radiation for patients with glioblastoma were conducted to combat the poor penetrance of the neutrons. However, without a sufficient tumour/normal boron concentration ratio, neutron irradiation led to similar rates of capture in healthy and tumour cells, leading to the death of healthy tissues and significant side effects for patients [41]. Due to the poor survival of patients and severe toxicities, initial BNCT studies were halted. In 1968, BNCT clinical trials for patients with high-grade gliomas proceeded due to the development of a new boron carrier, sodium mercaptoundecahydro-closo-dodecaborate, commonly known as BSH [42,43]. BSH has a high boron content with each molecule carrying twelve atoms, allowing for the delivery of large quantities of boron into cells. However, BSH diffuses across the cell membrane and is therefore not directed specifically to tumour cells, often yielding poor tumour/blood ratios of 0.2–0.4 in trials. Despite this, BSH has been shown to be useful for the treatment of malignant brain tumours that have a disrupted blood–brain barrier [44,45]. Indeed, some studies have reported ^10^B tumour/normal tissue ratios as high as 40 to 1 [3].

Following the development of BSH, the first tumour cell-specific boron carrier was developed, (L)-4-dihydroxy-borylphenylalanine, commonly known as BPA [46]. BPA is an analogue of the amino acid phenylalanine and is transported into cells via the L-type amino acid transporter LAT1. LAT1 is the most commonly upregulated and overexpressed LAT system transporter in a number of human cancers, such as head and neck cancers and where 61% of tongue cancers can display overexpressed LAT1 [47]. The upregulation of LAT1, and therefore an increase in amino acid transport, has been linked to the increased need of cancer cells for nutrients to facilitate increased cell proliferation and ultimately cancer progression. In support of this, studies have shown that inhibition of LAT1 with the LAT inhibitor 2-aminobicyclo-(2,2,1)-heptane-2-carboxylic acid (BCH) decreases cell viability by ~40–50% [48,49]. In 1987, clinical trials began treating malignant melanoma with BPA-mediated BNCT [46,50,51,52], and it was shown that BPA accumulates selectively in tumour cells, with tumour/blood ratios shown to be greater than 3 after six hours treatment with BPA [53]. Since the initial development of BSH and BPA, these boron carriers have been utilised in numerous clinical trials of different cancers, including glioblastoma and head and neck cancers. Historically, nuclear reactors have been employed as neutron sources for BNCT. However, the neutron energies generated typically allow for only limited penetration into tissues. To enhance neutron penetration depth, a recent clinical trial investigated the use of a cyclotron-based epithermal neutron source (C-BENS) in conjunction with ^10^B-condensed BPA (borofalan) as a boron delivery agent. This approach aimed to maximise the flux of thermal neutrons reaching deep-seated tumours. The findings demonstrated that this achieved an overall response rate of 71% in patients with recurrent squamous cell carcinoma and locally advanced or recurrent non-squamous cell carcinoma [54]. The details of completed or ongoing clinical trials are summarised in Table 1.

## 4. The Cellular Response to BNCT

### 4.1. Understanding the Efficacy of BNCT

The efficacy of BNCT has been examined over the last couple of decades both in vitro and in vivo, demonstrating the ability of this treatment to kill various tumour types. Focusing on in vitro, these studies are actually few in number. One study demonstrated that in WRO follicular thyroid carcinoma cells, BNCT mediated by BPA (10 µg ^10^B/mL) and an equivalent dose of tetrakis-carborane carboxylate ester of 2,4-bis (α,β-dihydroxyethyl)-deutero-porphyrin IX (BOPP) reduced tumour cell survival in a dose-dependent manner and to a greater extent than γ-radiation, with cell survival reduced by ~1.6–2-fold compared to radiation only [68]. Whilst not performed in vitro, a study using rat tumour graft models of lymphosarcoma examined the effect of BNCT on DNA damage [69]. Here, it was shown that there was an increase in the DSB marker γH2AX 6 h after BPA-mediated (330 mg/kg) BNCT compared to neutron irradiation alone. The persistence of DSB damage was also seen with approximately 40% of BPA-BNCT-treated cells positive for γH2AX 20 h post-BNCT treatment, compared to less than 5% in untreated controls.

As to experiments in vivo, substantially more evidence has been reported. An early in vivo study used an intracerebral rat gliosarcoma model to demonstrate the efficacy of BNCT [70]. Rats treated with two oral administrations of BPA (750 mg/kg) prior to BNCT had a 1-year survival rate of approximately 43%, whilst none of the untreated controls survived to 1 year. Interestingly, the study also demonstrated increased 1-year survival with dose fractionation, with two doses of BNCT resulting in an ~50% 1-year survival rate and three doses resulting in an ~60% 1-year survival rate, although these results were not statistically significant when compared to the single doses. Another early study utilised SCCVII mouse squamous cell carcinoma cells inoculated into the thighs of C3H/He mice to understand the effectiveness of BNCT mediated by BPA (1500 mg/kg), BSH (75 mg/kg) or a combination of the two drugs (750 mg/kg BPA and 75 mg/kg BSH) prior to neutron irradiation [71]. The results derived from cells obtained from the tumour tissue indicated that BPA-mediated BNCT cell survival was reduced by ~6-fold compared to treatment with combined BPA/BSH and more than 10-fold compared to BSH-mediated BNCT alone, which showed the least effect on cell survival. Using a mouse xenograft model of human prostate cancer, it has been demonstrated that BPA-mediated (250 mg/kg) BNCT resulted in a marked ~8-fold reduction in tumour size at 9 weeks post-treatment compared to the untreated controls [72]. The effectiveness of BNCT on lung metastases has been assessed utilising DHD/K12/TRb colon carcinoma cells injected into the jugular vein of rats, where either BPA or a combination of BPA and sodium decahydrodecaborate (GB-10), a boron carrier suspected to be distributed cellularly by diffusion, was utilised [73]. Both approaches demonstrated an ~1.4-fold increase in the survival time of the mice compared to the untreated controls, with no significant difference noted between the two BNCT applications. BNCT has also been tested on malignant gliomas in a study utilising mice subcutaneously implanted with U87 human glioblastoma cells, where BPA (350 mg/kg BPA), BSH (100 mg/kg) and liposomal BSH (100 mg/kg) were used to enhance the delivery of the drug [74]. In all treatments, BNCT slowed tumour growth compared to the untreated controls, with BPA and BSH demonstrating an ~2.5-fold reduction in tumour volume, whereas liposomal BSH was the most effective with a greater than 4-fold reduction. BNCT has additionally been utilised to treat dogs with head and neck tumours where no other therapeutic options were available [75]. Two applications of BNCT were delivered separated by 3–5 weeks with BPA–Fructose (BPA-F; 350 mg/kg) administered intravenously for 45 min prior to neutron irradiation. All five dogs treated exhibited a partial response, with a reduction in tumour volume ranging from 5 to 50% at 1–2 months post treatment. Survival ranged from 8.5 to 13.5 months post-treatment, exceeding the estimated survival time of 1–2 months. Only mild and reversible toxicities were noted in this study.

BNCT has furthermore been shown to be effective in preventing the development of tumours from precancerous lesions. Using the carcinogen 7,12-dimethylbenz[a]anthracene (DMBA) to induce precancerous lesions in the oral cavities of hamsters, these were subsequently treated with BNCT using BPA (15.5 mg B/kg), GB-10 (50 mg B/kg) or a combination of BPA and GB-10 (31 mg B/kg and 34.5 mg B/kg, respectively) [76]. All three BNCT protocols demonstrated significant inhibitory effects (ranging from 77 to 100%) on the development of tumours from precancerous lesions when compared to the mock-irradiated controls. However, the effects of GB-10-mediated BNCT were transient, and a similar effect was seen with neutron irradiation alone. The inhibitory effects of BPA and BPA plus GB-10-mediated BNCT were more persistent, with both treatment protocols maintaining a 51% inhibition rate at 8 months post BNCT treatment.

Collectively, these in vitro and in vivo studies clearly demonstrate the tumour cell killing effects of BNCT whilst minimising effects to the normal tissues and enhancing survival. Minimal toxicities have been reported in vivo, with side effects often resolving once the treatment is complete. The evaluation of these data, in conjunction with multiple clinical trials, demonstrates that BNCT is an effective therapeutic modality, potentially for a wide range of cancer types.

### 4.2. BNCT Under Hypoxic Conditions

One of the advantages of BNCT over conventional X-ray radiotherapy is the ability to treat hypoxic cell populations, as there is a much-reduced requirement for oxygen to enhance its effects due to the high-LET ions generated. Interestingly, although neutron capture reactions can occur in low oxygen conditions, a number of studies have demonstrated a limited uptake of BPA in both hypoxic and quiescent cell populations and a subsequent reduction in the efficacy of BNCT in these populations compared to proliferating cells. One study utilising B16-BL6 melanoma tumour-bearing mice showed that the total tumour cell population exhibited a higher sensitivity to BNCT than the quiescent cell population, demonstrating an ~1.4-fold increase in micronuclei formation following BPA-BNCT and an ~1.6-fold increase following BSH-BNCT [77]. Furthermore, *lat1* mRNA expression has been shown to be downregulated under conditions of reduced oxygen (1% and 10%), resulting in up to an ~7.5-fold decrease in boron uptake [78]. Hypoxia-inducible factor 1α (HIF-1α) is expressed by cells in response to hypoxic conditions and mediates a broad range of cellular responses to low-oxygen conditions. Interestingly, HIF-1α has been shown to affect LAT1 expression in several different cancer cell lines. Indeed, it has been observed in T98G glioblastoma, HSC-3 oral squamous cell carcinoma and MCF-7 breast adenocarcinoma cells that hypoxic conditions lead to an ~31–48% decrease in *lat1* mRNA expression and that the expression could be restored by using an siRNA knockdown of *hif-1α* [79]. In the same study, it was demonstrated that across the three cell lines, hypoxia reduced the cellular sensitivity to BPA-mediated BNCT, resulting in a 1.25–1.7-fold increase in the cell survival fraction. One study showed that the proportion of glioblastoma and metastatic tumour cells expressing LAT1 was 3 times higher than the proportion expressing proliferating cell nuclear antigen (PCNA), a marker of cell proliferation, suggesting that quiescent cell populations, such as those that occupy hypoxic tumour regions, do in fact express LAT1 [80]. Nevertheless, more studies are needed to understand the impact of mild and severe hypoxia (including chronic exposure) on LAT1 expression, BPA uptake and, therefore, the effectiveness of BNCT in comparison to normoxic conditions.

### 4.3. Improving BNCT with Combinational Therapies

Several combination therapies with BNCT have been proposed in an effort to improve boron carrier uptake and therefore increase the effectiveness of the treatment in killing cancer cells.

#### 4.3.1. Combining Boron-Containing Compounds

A few studies have suggested that BNCT can be improved by combining different boron carriers. By combining BPA with GB-10, it was shown in hamster cheek oral cancers that this combination treatment resulted in an ~2-fold increase in mean tumour boron concentrations and a 3.3-fold increase in homogeneity compared to BPA treatment alone [81]. Reductions in tumour volume have also been observed in a rat colon cancer model by combining BPA administration with the intravenous injection of GB-10, where the incidence of tumours shrinking to ≤50% of their original size increased by ~2-fold with the combination treatment compared to BPA alone [82].

#### 4.3.2. Improvement in Boron Cellular Uptake

Another strategy for enhancing BNCT effectiveness is combining with other types of drugs and treatment modalities to increase boron uptake. One study utilising a rat brain tumour model showed that the intracarotid administration of Cereport, a drug that increases blood–brain barrier permeability, along with the intracarotid administration of BPA increased the uptake of BPA by ~2-fold into tumour cells [83]. Another study [84] utilised a hamster cheek pouch oral cancer model to demonstrate that the uptake of GB-10 in tumours could be enhanced ~2.5-fold with electroporation involving short electronic pulses applied to the tumour tissue to increase cell permeability. Combining GB-10 administration with electroporation prior to neutron irradiation has also been shown to improve overall tumour responses compared to GB-10 alone in the hamster check pouch oral cancer model. Here, an overall tumour response, including partial and complete remission, of 92% was achieved with GB-10 combined with electroporation compared to 48% for GB-10 administration alone [85]. It has furthermore been shown in B16-BL6 melanoma tumour-bearing mice that BPA-mediated BNCT can be enhanced with mild-temperature hyperthermia (MTH) and nicotinamide, an acute hypoxia-releasing agent. Surviving fraction enhancement ratios of 1.2–1.4 were seen with MTH and 1.15–1.25 with nicotinamide. The frequency of micronuclei was additionally increased with these combination treatments, with enhancement ratios of 1.2–1.3 with MTH treatment and 1.15 with nicotinamide treatment [77,86]. Further studies by this group found that BNCT combined with MTH and tirapazamine, a toxic radical activated in low oxygen conditions, was able to enhance local tumour control, resulting in an enhancement ratio of 2.2, whilst slightly supressing the formation of distant lung metastases, likely by improving the effect of BNCT on the hypoxic cell population [87]. Another study tested the HIF-1α inhibitor YC-1 in combination with BNCT and showed that this was able to sensitise hypoxic tumour glioblastoma and oral squamous cell carcinoma cells to the effects of BPA-mediated BNCT by ~1.7- and ~3.4-fold, respectively, possibly due to an increased expression in LAT1 following HIF-1α inhibition, enabling a greater uptake of BPA [79].

#### 4.3.3. Suppression of the Tumour Immune Response

Mediating the immune response is another potential strategy to enhance the effects of BNCT. Myeloid-derived suppressor cells (MDSCs) are negative regulators of immune function, and their presence in the tumour environment is associated with tumour progression. It has been shown that although BNCT led to an initial decrease in the infiltrated MDSCs, a subsequent ~3-fold increase in MDSCs circulating in the blood 14–21 days post-treatment was observed, which may hinder the efficacy of the treatment [88]. Treatment with PLX-3397, an inhibitor of CSF-1R which is required for MDSC maturation and tumour localisation, was demonstrated to inhibit the accumulation of MDSCs following BNCT, and an ~3.5-fold increase in tumour infiltrating CD8+ T cells was noted compared to BNCT alone, enhancing the immune response to the tumour. Furthermore, mice with oral tumours treated with PLX-3397 and BNCT exhibited prolonged survival (90 ± 26 days) compared to those treated with BNCT alone (66 ± 16 days). Another study combined BPA-mediated BNCT with the immune stimulant Bacillus Calmette–Guérin (BCG) in an ectopic colon cancer model in rats [89]. Adult BDIX rats, injected subcutaneously with DHD/K12/TRb colon cancer cells to establish tumour growth, received three doses of BCG by injection, and BNCT was performed in between the first and second drug doses. Two weeks post-irradiation, the rats were injected subcutaneously with colon cancer cells in the left hind flank. Both BNCT and BNCT with BCG induced significant local anti-tumour responses in the right flank tumours, but there were no significant differences between the two treatments. Both treatment protocols contributed to a regional effect, reducing the tumour volume in distant tumours. Whilst 10% of the animals treated with BNCT showed a significant shrinkage of the tumour volume compared to the controls, the reduction reached 38% for the combination with BCG. These results indicate that BNCT can possibly lead to an abscopal effect in the regression of distantly formed tumours.

## 5. New Boron Delivery Agents

As well as combining different treatment modalities, efforts to improve the effectiveness of BNCT have focused on the development of new boron compounds. Although BSH and BPA have been used clinically due to their low toxicity, accumulating sufficient boron specifically in tumour cells needed to create optimal BNCT effectiveness can be difficult, with success varying between different studies. In this section, we will discuss four groups of new boron compounds that aim to improve boron delivery to tumours, particularly focusing on those that have been tested in vitro and in vivo. As a more comprehensive view of developments over the last two years, a summary of the newly studied boron delivery agents evaluated for their toxicity, biodistribution, and efficacy in BNCT in vivo, is included (Table 2).

One approach to improving boron delivery is to increase the effectiveness of the clinical boron compounds BSH and BPA through conjugation with other molecules.

### 5.1. BSH Derivatives

Since BSH does not target tumour cells specifically, several BSH-conjugated compounds have been synthesised to improve tumour localisation. Porphyrins, such as chlorins, have been shown to have high tumour uptake whilst maintaining low toxicity [125]. One study synthesised BSH-conjugated chlorin derivatives, demonstrating that two compounds exhibited good tumour-selective accumulation in colon tumour-bearing mice, reaching a maximum of ~0.2 pmol/mg [126]. Another study developed a boron cluster combining BSH with the cell-penetrant lipopeptide pepducin, which demonstrated enhanced cellular uptake (<200 ng B/10^6^ cells) and decreased survival of T98G glioblastoma cells when combined with neutron irradiation compared to BSH alone which resulted in undetectable intracellular boron levels and was not able to increase the sensitivity of cells to neutron irradiation [127]. Poly-arginine peptide-conjugated BSH (polyR)(BSH-polyR) was observed to be able to penetrate the plasma membrane of various glioma, breast cancer and pancreatic cancer cell lines, with uptake correlated to CD44 expression, suggesting CD44 glycoprotein as a cellular target of the compound [128]. BNCT was conducted using one BSH-polyR compound, termed BSH-11R, in two glioblastoma cell lines with different CD44 expression. This showed that MGG4 cells with low CD44 expression showed only a 10% difference in survival between cells treated with or without BSH-polyR prior to neutron irradiation, whereas the high-CD44-expressing MGG18 cells exhibited more than 80% cell death with the BSH-11R treatment and neutron irradiation.

One study synthesised A6K peptide nanotubes as a BSH delivery system and showed significantly improved delivery of boron to U87 glioma cells 24 h after treatment, where the intracellular boron concentrations were approximately ten times greater than those achieved with BSH alone [129]. In vivo studies using a mouse glioma brain tumour model also indicated that the A6K/BSH complex resulted in a higher accumulation of boron in the tumour tissue compared to BSH alone, where this was >10-fold at the highest concentration of A6K (400 µM). This study also monitored cell survival with A6K/BSH complex (200 µM)-mediated BNCT, which resulted in an ~3-fold decreased survival of tumour cells compared to BSH-mediated BNCT. Importantly A6K peptide nanotubes appeared to show no evidence of toxicity, indicating the relative safety of the drug. Recently, organosilica nanoparticles containing BSH have been developed. In OVCAR8 human ovarian cancer cells, BSH-BPMO nanoparticles demonstrated an ~50-fold increased uptake of boron after 24 h of treatment compared to BPA, which was additionally and significantly more effective than BSH [130]. Fluorescent microscopy showed that BSH-BPMO nanoparticles labelled with Rhodamine b accumulated in the perinuclear region and demonstrated good distribution throughout OVCAR8 spheroids. BNCT irradiations performed on OVCAR8 spheroids revealed that BSH-BPMO-treated spheroids were completely destroyed following neutron irradiation, compared to BNCT irradiation with either BSH or BPA where spheroid sizes were only reduced to ~60% and ~45%, respectively, of the size of the unirradiated controls.

### 5.2. BPA Derivatives

BPA is accumulated in tumour cells through the amino acid transporter LAT1 which is often upregulated in tumours. However, and as previously discussed, studies have shown BPA uptake in hypoxic and quiescent cell populations to be limited, and boron tumour/normal and tumour/blood ratios in some studies have been insufficient to ensure minimal toxicity to healthy tissues. A specialised drug delivery system designed to enhance the uptake of BPA into tumour cells, called ‘Bioshuttle’-p-BPA_10_, has been developed consisting of BPA covalently linked to a peptide containing a nuclear localisation sequence (NLS) [131]. A 1 h incubation with ‘Bioshuttle’-p-BPA_10_ demonstrated nuclear boron concentrations of ~0.2 µg/g after 24–72 h, although no direct comparisons to BPA only or of any BNCT irradiations were included in this study. Another study utilised dipeptides of BPA and tyrosine (BPA-Tyr and Tyr-BPA) to enhance the selective delivery of the compound into tumour cells, specifically via the oligopeptide transporter PEPT1 which is often upregulated in tumours [132]. These dipeptides provide an alternative entry route into tumour cells independent of LAT1. It was shown that an siRNA knockdown of PEPT1 in AsPC-1 pancreatic cells resulted in ~2-fold reduced cellular boron concentrations when cells were treated with both BPA-Tyr and Tyr-BPA, indicating that PEPT1 is a primary transporter of the dipeptides into cells. This study also demonstrated that in mice bearing AsPC-1 xenograft tumours treated intravenously with BPA-Tyr (370 mg/kg), tumour boron concentrations at 3 and 5 h were greater than blood boron concentrations, resulting in tumour/blood ratios of ~2.0.

Recently, it has been described that poly(vinyl alcohol) (PVA) can form complexes with BPA in solution and that PVA-BPA complexes have enhanced cellular uptake compared to BPA as well as reduced efflux from the cytosol [133]. In vitro studies in BxPC-3 human pancreatic adenocarcinoma cells showed that PVA-BPA achieved intracellular boron concentrations of 6.4% and 6.8% dose/g tumour 1 h and 6 h, respectively, after treatment. In comparison, BPA-F treatment yielded a maximum boron concentration of 3.9% and 1.2% dose/g tumour 1 h and 6 h, respectively, after treatment. These results indicate that PVA-BPA complexes can accumulate more boron into cells that is retained over time compared to BPA-F. This study also showed in both subcutaneous hypovascular BxPC-3 and CT26 mouse tumour models that BNCT mediated by PVA-BPA reduced tumour volume to a significantly greater extent (~4-fold) than BPA-F, although both were shown to be internalised by LAT1.

### 5.3. Boronated Compounds

Another approach to developing new boron carriers for BNCT focuses on boronated compounds including amino acids, peptides and carboranes.

#### 5.3.1. Amino Acids

Recently, 3-borono-L-tyrosine (BTS) has been developed, and boron uptake in FaDu head and neck cancer cells after 24 h was shown to reach a maximum concentration of ~2000 µg boron/g protein, compared to ~1250 µg boron/g protein achieved following BPA treatment [93]. Drug retention experiments showed that BTS had an ~2-fold greater retention in cells compared to BPA at 2 and 18 h post-removal of the drug-containing media. In vivo studies utilising FaDu xenograft mouse models also demonstrated a greater retention of BTS, where boron concentrations between 1 and 4 h post-treatment were reduced by 43% in BTS-treated mice compared to 57% for those treated with BPA. Despite this, no BNCT efficacy studies relating to tumour cell death or tumour growth control were performed.

#### 5.3.2. Peptides

Drug designs utilising peptides have demonstrated good selectivity and safety in the treatment of a number of diseases; thus, several studies have aimed to utilise peptides as effective boron carriers for BNCT. One study synthesised Tyr^3^-octreotate (TATE) derivatives to improve the tumour selectivity of boron uptake, particularly in neuroendocrine tumours that overexpress somatostatin receptors [134]. Binding assays performed in CHO-K1 Chinese hamster ovary cells and CCL39 Chinese hamster fibroblast cells expressing human somatostatin receptor subtypes showed that closo-borane-conjugated TATE derivatives were capable of binding to the various receptor subtypes to varying degrees, offering an alternative entry into tumour cells aside from LAT1. Another study utilised cell-penetrating peptide (CPP)-conjugated dodecaborate compounds for in vitro BNCT studies, which can deliver up to 12 boron atoms [135]. One compound, designated RB-*RLA*, utilised the arginine-rich RLA peptide designed to enhance mitochondrial targeting and retention in cells, which demonstrated superior (~3.7-fold) uptake in C6 glioma cells compared to BSH after a 30 min treatment. The surviving fraction of cells treated with 5 µM DB-*RLA*-mediated BNCT was reduced by up to 50% compared to cells treated with 1 mM BSH-mediated BNCT, indicating increased effectiveness.

More recently, several boron–peptide conjugates have been developed; in particular, the compound ANG-B was synthesised from a modified sequence of angiopep-2, a common drug delivery ligand capable of crossing the blood–brain barrier, conjugated to ^10^B-4-carboxyphenylboronic acid [95]. A comparison between BPA and ANG-B uptake was conducted in several cancer cell lines, where U251 glioma and A549 non-small-cell lung cancer cells demonstrated similar boron uptake with both drugs, whereas U87MG glioma cells accumulated slightly more boron with BPA treatment. However, the ANG-B treatment showed an increased accumulation of boron in HS683 glioma, A357 melanoma and HepG2 hepatocellular carcinoma cells, between 2- and 3-fold more than with using BPA. Subsequently, it was shown in HS683 cells that ANG-B-mediated BNCT resulted in a significantly greater reduction (~1.2-fold) in cell survival compared to BPA-mediated BNCT. ANG-B also demonstrated significantly greater tumour/normal (~5.6) and tumour/blood (~9.4) ratios compared to BPA (<2.5) 4 h after injection into the tail vein of intracranial HS683 glioma-bearing mice. Additionally, tumours treated with ANG-B (2.5 mM/kg)-mediated BNCT showed a significant ~2.2-fold reduction in volume 31 days after treatment, compared to BPA-mediated BNCT.

#### 5.3.3. Carboranes and Hybrid Dual-Action Compounds

Due to their high boron content, carboranes offer a potential basis for new boron compound development. One study developed agonist conjugates based on the selective carborane-functionalised gastrin-releasing peptide receptor (GRPR), given that this is overexpressed in various malignant tissues [136]. This study successfully introduced multiple bis-deoxygalactosyl-carborane building blocks to a GRPR-selective ligand, resulting in conjugates carrying up to 80 boron atoms per molecule, which demonstrated successful internalisation into PC3 prostate cancer cells that express the GRPR. More recently, a study synthesised two radioactive boron compounds, [^67^Ga]16 and [^125^I]17, required for biodistribution measurements and conjugated a closo-dodecaborate moiety ([B_12_H_12_]^2−^) with a gallium–DORA-c(RGDfK) complex, which contained an RGD peptide for tumour targeting [119]. Cellular uptake studies in U87 MG glioblastoma cells showed the time-dependent accumulation of both drugs over a 6 h period. The biodistribution of both drugs was assessed in U87 MG tumour-bearing mice showing the accumulation of the drugs in tumour tissues, particularly [^125^I]17, which demonstrated ~8.5% of the injected dose/g in the tumour after 4 h. However, high accumulation was also noted in other tissues, most notably the kidneys (~8.9%) and large intestine (~7.4%), 4 h post-injection. The co-injection of [^67^Ga]16 and [^125^I]17 with an excess of c(RGDfK) peptide which binds and competes for αvβ3 integrin resulted in the reduced accumulation of both drug complexes in the tumours of U87 MG tumour-bearing mice, indicating that the complex uptake is αvβ3 integrin-dependent. Recently, new carborane-bearing hydroxamate matrix metalloproteinase (MMP) ligands have been synthesised for BNCT, which showed in vitro that the biological effectiveness in squamous cell carcinoma SCCVII cells and glioma U87 delta EGFR cells was up to three times more than with BPA [137]. Interestingly, this study showed BNCT efficacy through binding to the overexpressed surface receptor, bypassing tumour cell uptake. Hybrid dual-action compounds (tyrosine kinase receptors inhibitor and closo-carboranyl and metallacarboranyl moieties) reduced the cell survival in HT-29 cells after neutron irradiation by 6-fold more than with BPA [138]. Additionally, F98 astrocyte cells treated with the hybrid compound at 10 μM for 4 h showed an ~5-fold increased uptake compared to BPA at 0.925 mM, without any significant decrease over time as a result of efflux by extracellular amino acid exchange which was observed for BPA. Hybrid dual-action compounds targeting G-coupled proteins (overexpressed in cancer cells) also show promising results in cellular uptake. Peptides generated by linking the gastrin-releasing peptide receptor (GRPR)-selective ligand to multiple bis-deoxygalactosyl-carborane building blocks showed selectivity for PC3 cancer cells but not for HepG2 cells, suggesting a reduced uptake into liver cells [136]. In HEK293 cells, short, boron-rich meta-carborane conjugates that target the ghrelin receptor demonstrated successful cellular internalisation of the compounds, as well as high stability of the meta-carborane building block [139].

### 5.4. Nanoparticles

The use of nanoparticles as boron carriers for BNCT has been studied in recent years. One study synthesised poly(DL-lactide-co-glycolide) (PGLA) and poly(L-lactide-co-glycolide) (PLLGA) nanoparticles of 100 or 150 nm in size loaded with o-carborane and studied their biodistribution [140]. Data acquired in B16 melanoma tumour-bearing mice injected with the drug into the tail vein showed good accumulation of boron in the tumour, with reduced accumulation in tissues and the blood. The 100 nm PLLGA nanoparticles resulted in the greatest boron accumulation in the tumour (~114 µg/g tissue) after 8 h of treatment. For all the nanoparticles tested, the tumour/blood boron concentrations ratios exceeded 5 at both 8 h and 12 h post-administration. Another study developed boron carbon oxynitride (BCNO) nanoparticles, either unmodified or functionalised with polyethylene glycol (PEG) polymers (to enhance solubility and stability) or with polyethyleneimine (PEI) (to promote drug uptake) [141]. After 4 h of incubation, the uptake of bare BCNO nanoparticles into ALTS1C1 murine anaplastic astrocytoma cells was negligible, compared to the PEG and PEI nanoparticles which achieved intracellular boron concentrations of ~16 µg and 48 µg, respectively. The viability of ALTS1C1 cells 6 h after treatment with PEG or PEI nanoparticle (25 µg/mL)-mediated BNCT and neutron irradiation was reduced by 37% and 43%, respectively, compared to the unirradiated controls, whereas BPA-mediated BNCT only resulted in a 13% reduction in cell viability. A separate study utilised boron nitride nanoparticles (BNNPs) as boron carriers which can undergo rapid degradation under physiological conditions [142]. The BNNPs were engineered with a phase-transitioned lysozyme (PTL) coating to protect the nanoparticles from hydrolysis when circulating in the bloodstream. After 12 h of treatment, a 125 ppm intracellular boron concentration was detectable in 4T1 murine breast cancer cells, which was replicated in 4T1 tumour-bearing mice 24 h post-treatment. However, significant boron accumulation occurred in the liver, surpassing the concentration achieved in the tumour. In response to PTL-BNNP-mediated BNCT, a significantly reduced tumour volume compared to unirradiated controls was observed (325 mm^3^ and 1485 mm^3^, respectively), and increased numbers of mice survived at 21 days post-irradiation. To examine boron clearance from the liver and spleen of 4T1 tumour-bearing mice post-neutron irradiation, vitamin C was used to detach the PTL coating from the nanoparticles, which showed the clearance of boron from both organs compared to mock-treated mice.

### 5.5. Antibody-Based Compounds and Targeted Therapies

Antibody-based drugs are a class of therapeutic drugs that have become increasingly prevalent in recent years. The overexpression of epidermal growth factor receptor (EGFR) in several cancer types, including glioblastoma and head and neck cancers, makes it an attractive target for therapeutic intervention. Cetuximab, a monoclonal antibody targeting EGFR, has been used to develop a novel boron carrier for BNCT [143]. Boronated cetuximab (C22-G5-B_100_) was administered directly to the brain of F98 glioma-bearing rats, which showed tumour boron concentrations increased by ~2.5-fold in EGRF-positive versus wild-type tumours, with undetectable levels of boron observed in the blood, liver, kidneys and spleen. In BNCT irradiations using either BPA (500 g/kg) or C22-G5-B_100_/CED containing either 30 or 40 µg of boron, both showed marginally improved survival when exposed to neutron irradiation compared to untreated controls, while the combination of the two drugs yielded a significantly increased survival rate of up to 59 days. More recently, a study has targeted HER-2, a protein overexpressed in some breast cancers, using an antibody conjugated to a boron nitride nanotube/β-1,3-glucan-IgG complex (BNNT/β-glucan-IgG complex) [101]. In SK-OV3 xenograft mouse models, it was shown that the intravenous injection of the BNNT/β-glucan-IgG complex resulted in boron accumulation in tumours to a greater extent than in other tissues, reaching an average concentration of ~370 µg/g tissue. The tumour/blood and tumour/normal ratios reached 390 and 170 at 24 h post injection, indicating the high tumour-specific accumulation of the drug. Finally, another recent study successfully developed a computational pipeline for designing boron delivery antibodies that can enhance the efficacy of BNCT [144], although the biological applicability of this has yet to be tested. Pentagamaboronon (PGB-0), a curcumin analogue containing a boronic acid group, has been developed as a chemotherapeutic agent and boron carrier specifically targeting HER-2-overexpressing tumours [145]. Curcumin is a well-documented inhibitor of tyrosine kinases, including HER-2. In studies, PGB-0 exhibited cytotoxic effects on HER-2-overexpressing breast cancer cells, with an IC_50_ value of 270 μM, while demonstrating no detectable toxicity in normal cell lines. Furthermore, treatment with 150 μM PGB-0 led to a significant reduction in HER-2 expression in breast cancer cells. Although PGB-0 has shown promise as a standalone therapeutic agent, its potential utility as a boron carrier for BNCT has not yet been established. Further studies are required to evaluate boron distribution and the effects with neutron irradiation to fully assess its potential in BNCT applications.

## 6. Discussion

BNCT holds promise in being able to more effectively treat tumours that are inherently resistant (such as solid, hypoxic tumours) or those that acquire resistance to conventional radiotherapy using X-rays. The tumour-specific targeting of BNCT is designed to minimise radiation exposure to normal tissues, thereby enabling the treatment of patients who are unable to receive further doses of conventional radiotherapy due to the risk of radiation-induced side effects. Due to the production of high-LET helium and lithium particles generated during the neutron capture reaction, BNCT has been demonstrated to be more biologically effective than conventional radiotherapy in promoting tumour cell killing, with the potential to address treatment resistant cancers, as evidenced by varying degrees of success in clinical trials. As a treatment for cancer, BNCT has some areas of common ground with targeted radionuclide therapy (TRT) or perhaps most especially with targeted alpha therapy (TAT) (see [146,147] for comprehensive and recent reviews). These treatment approaches are rapidly expanding as new agents become available and evidence mounts of their benefits for patients. Clinically, ^223^Ra is the most commonly used alpha-emitting isotope, with the main energies of emitted alpha particles in the range of 5.8 to 7.5 MeV, corresponding to ranges of approximately 35–75 μm in tissue. These particle ranges are much larger than those associated with the emissions from BNCT (~12–14 μm), meaning that there is an increased probability of the dose being spread to adjacent cells from those cells targeted with ^223^Ra. This can be advantageous if the adjacent cells are tumour cells, rather than normal, healthy cells. In contrast, targeting every tumour cell is much more important in BNCT, and this is one of the great challenges for the field. Clearly, BNCT benefits greatly from the fact that the boron carrier compounds are intended to be non-toxic so the risk of damage to other important body tissues (liver, kidney, etc.) is much reduced compared to any therapy that involves the administration of a radioactive compound.

The biological effectiveness of BNCT is well established to be driven through the formation of more complex and irreparable DNA damage that persists following treatment. Despite this, surprisingly, the precise nature of the DNA damage induced and the cellular response to this after BNCT treatment is not well understood. There are relatively few in vitro studies that have been performed investigating this, but these have already identified potential conflicts in the repair pathways and proteins that are involved in the resolution of the damage induced by BNCT. Indeed, there is still confusion as to whether there is a reliance on HR for the repair of BNCT-induced CDD driven by proteins such as MRN, BRCA1/2 and RAD51, whereas others propose roles for NHEJ involving either DNA-PKcs, Ku70/80 or PARP-1. With the lack of this evidence, more systematic in vitro studies using well-characterised 2D and 3D tumour cell models and neutron irradiation conditions are required to more comprehensively understand the DNA damage profile relative to boron uptake and the proteins that are essential for co-ordinating the repair response. This could also lead to the potential for targeting these proteins and pathways to further improve the biological effectiveness of BNCT.

The biological effectiveness of a compound used in BNCT is a complex function of boron distribution and its heterogeneity between and within tumour cells, along with the biological effects of the high-LET reaction products. This has been long understood, and many studies have attempted to assess boron microdistribution from an experimental perspective. Important work in this area includes the use of atomic absorption spectroscopy to analyse subcellular compartments [148] and the use of ion mass spectrometry to image boron distribution in cells [149]. There have also been important efforts to understand this dependency from a computational standpoint ranging from early work [150] to more recent approaches that focus on the prediction of the RBE [151,152,153]. These efforts are moving the field towards a capability, via simulation, to predict the biological effects of boron compounds with different subcellular boron distributions. In the context of experimental cell biology, a recent dosimetry study has helped to illuminate the dependence of the dose delivered to cells in a monolayer on the substrate material on which the cells are grown [154]. All of these simulation efforts are valuable for benchmarking and improving understanding from a theoretical perspective and apply very well to the model systems commonly used in in vitro and in vivo experimentation. However, mature human tumours have been shown to exhibit a very high degree of heterogeneity in their cellular make-up [155]. Such pathological analysis is helpful in defining the next challenge for BNCT modelling studies to begin to address the wide variety of tumour cell types within a single tumour. As a further layer of complexity, a large and variable part of the tumour mass is comprised of immune cells. The potential to spare these immune cells in a targeted treatment such as BNCT, combined withthe potential of a high-LET dose to initiate an immune response and for this to be further enhanced with combination treatment with immune checkpoint inhibitors (illustrated by [156]), could impact greatly on the biological response of a patient after therapy. In the context of the modelling of biological effects, this level of complexity is very challenging and is yet to be comprehensively addressed.

Clinical trials utilising the historic boron carriers, BSH and BPA, have demonstrated the ability to reduce tumour growth and prolong the survival of patients. However, a lack of consistency in drug dosing and neutron irradiation protocols between clinical trials makes it difficult to determine the optimum treatment protocol for patients. Furthermore, the penetration depth of epithermal neutron beams is restricted to ~6–8 cm, considerably less than that of photon beams, constraining the effectiveness of BNCT to tumours that are not deep-seated [157]. Additionally, variability in the expression of the LAT1 transporter across tumours, particularly in hypoxic regions, restricts the targetable tumour cell populations when using current boron carriers such as BPA. Incomplete irradiation of all tumour cell populations allows for the persistence of treatment-resistant clones, ultimately resulting in tumour recurrence. Due to the insufficiency of BSH (and to a lesser extent BPA) to accumulate boron levels specifically and in high quantities in tumour cells, the development of new boron compounds has been a primarily focus of recent BNCT research. In this review, we covered four of the most common types of new boron carriers; however, there are an increasing number of alternative carriers being investigated (covered in further detail here [158]). Radiotherapy primarily aims to induce irreparable DNA damage, and the high-LET particles generated by the BNCT reaction are known to cause increased CDD. Consequently, boron carriers that localise in proximity to the nucleus, such as boron-conjugated nucleic acids, are often prioritised in drug development to increase the probability of directly damaging the DNA. For a boron carrier to be effective in BNCT, it must achieve and maintain high tumour/normal tissue and tumour/blood ratios to minimise damage to healthy tissues. Thus, boron carriers that selectively target tumour-specific factors, such as overexpressed receptors, are preferred for improving therapeutic outcomes. Likely due to a lack of access to neutron facilities, newly synthesised compounds are rarely tested both in vitro and in vivo in combination with neutron radiation, limiting our understanding of how these compounds accumulate in different tissues and importantly how these compounds compare to BSH or BPA in BNCT. This ultimately has slowed down our ability to understand the optimal therapeutic strategies using BNCT and ultimately improve the outcomes of patients with aggressive and radioresistant tumours treated with BNCT using these compounds.

In summary, evidence demonstrates the significant ability of BNCT to more effectively kill tumour cells than conventional radiotherapy whilst limiting radiation-induced side effects to the normal tissues and organs at risk. Understanding the spectrum of DNA damage and the pathways co-ordinating repair in response to BNCT opens the option for combination treatment with DNA repair inhibitors, as well as improving drug formulations, including importantly altered formulations of BPA. A deeper understanding of the cellular mechanisms responding to BNCT, plus the development of improved boron carriers providing specificity to the tumour, will unquestionably lead to more sustained clinical trials and the future utilisation of this modality for the benefit of cancer patients.

## Figures and Tables

**Figure 1 cells-13-02065-f001:**
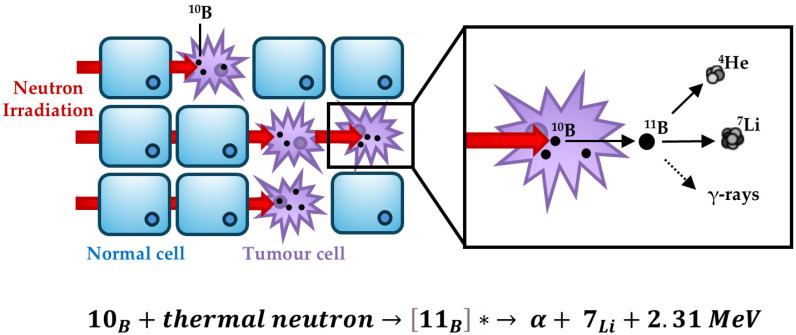
Overview of BNCT. The specificity of BNCT is driven by the uptake of boron (^10^B) into the cancer cells. Thermal neutron irradiation results in neutron capture by ^10^B to produce ^11^B (* represents an excited state), this then generates high-LET helium and lithium ions that cause extensive and localised DNA damage (CDD) within the tumour cells causing their death, whereas the normal cells remain intact.

**Figure 2 cells-13-02065-f002:**
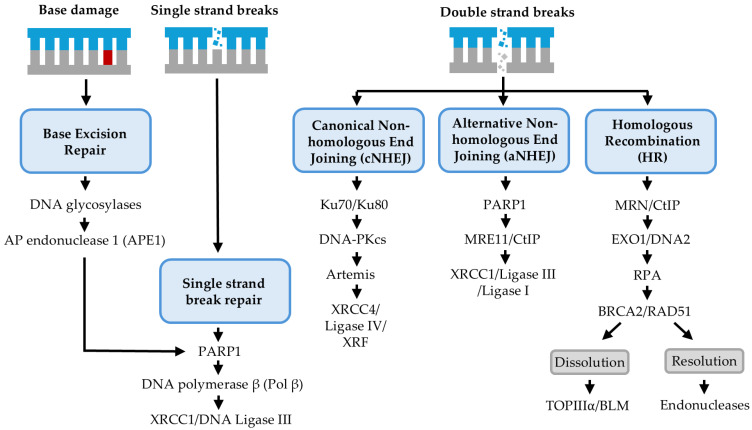
The cellular DNA damage response. Ionising radiation can generate multiple DNA lesions, including DNA base damage and SSBs that are repaired by proteins of the base excision repair (BER) pathway, whereas DSBs are repaired by non-homologous end joining (NHEJ) and homologous recombination (HR). During BER, base damage is recognised and removed by damage-specific DNA glycosylases in concert with APE1, generating a SSB as an intermediate. SSBs are recognised by PARP-1 and subsequently repaired by the polymerase and ligase actions of Pol β and XRCC1/DNA ligase IIIα, respectively. DSBs are usually repaired by either NHEJ (cNHEJ or aNHEJ) in the G_1_ phase of the cell cycle, whereas in S/G_2_, these are repaired by HR. In cNHEJ, Ku70/80 heterodimers bind to the DSB ends, enabling recruitment of DNA-PKcs and other end-processing proteins, such as Artemis. Ligation is performed by XRCC4-DNA ligase IV-XLF. In aNHEJ, PARP-1 binds to the DSB ends, and then limited end resection is performed by the MRN complex with CtIP. Where necessary, DNA synthesis is performed before ligation by either DNA ligase I or XRCC1-DNA ligase III**α**. During HR, and when a sister chromatid is available, extensive end resection is performed by the MRN complex and CtIP which results in the generation of ssDNA. The ssDNA is then coated by RPA, replaced by RAD51 to form nucleofilaments, and these undergo homology search and invasion of the sister chromatid, facilitated by BRCA2. DNA synthesis is completed, and then the dissolution or resolution of Holliday junctions is performed.

**Table 1 cells-13-02065-t001:** Clinical trials utilising BSH and/or BPA.

Tumour Type	Boron Carrier	Dates	Country	Drug Treatment Schedule	Blood Boron Concentration	Trial ID	Reference
Glioblastoma	BPA-F	1999–2001	Finland	290–400 mg/kg for 2 h	-	NCT00115440/53	[55]
Glioblastoma	BSH	2000–2002	Czech Republic	100 mg/kg for 1 h	-	NA	[56]
Glioblastoma	BPA	2001–2003	Sweden	900 mg/kg for 6 h	24 μg/g (15–32 μg/g)	NA	[57]
Glioblastoma	BPA-F	2000–2003	Sweden	900 mg/kg for 6 h	24.7 mg/g (range 15.2–33.7 mg/g)	NA	[58]
Malignant gliomas	BSH and BPA	2002–2003	Japan	BSH: 5 g total for 1 h (12 h before BNCT); BPA: 250 mg/kg for 1 h (1 h before BNCT)	-	NA	[59]
Malignant gliomas	BSH	1999–2002	Japan	100 mg/kg for 1–1.5 h	29.9 (18.8–39.5 μg/g)	NA	[60]
Malignant gliomas	BSH BPA	1998–2007	Japan	BSH: 5 g total for 1 h (12 h before BNCT); BPA: 250 mg/kg for 1 h (1.5 h before BNCT)	BSH: 32.9 ± 12.2 μg/g (protocol 1); 34.6 ± 9.6 μg/g (protocol 2); BPA: 17.4 ± 2.4 μg/g	NA	[61]
Malignant gliomas	BPA-F	2001–2008	Finland	290 mg/kg for 2 h; 450 mg/kg for 2 h	14 μg/g; 18 μg/g	NCT00974987	[62]
Malignant gliomas	BPA	2002–2007	Japan	Protocol 1: 100 mg/kg of BSH and 250 mg/kg of BPA for 1 h; Protocol 2: 100 mg/kg of BSH for 1 h and 700 mg/kg of BPA for 6 h	-	NA	[63]
Head and neck	BSH and BPA	2004–2007	The Netherlands	BSH: 50 mg/kg for 1 h; BPA: 100 mg/kg; BPA for 2 h	After BSH infusion, the ^10^B-concentration ratio of tumour/blood was 1.2 ± 0.4; after BPA infusion, the ^10^B-concentration ratio of tumour/blood was 4.0 ± 1.7	NCT00062348	[64]
Head and neck	BPA-F	2003–2012	Finland	400 mg/kg for 2 h	1st irradiation: 19.6 μg/g (13.0–26.5 μg/g); 2nd irradiation: 16.4 μg/g (10.9–23.0 μg/g)	NCT00927147 NCT00114790	[65]
Head and neck	BPA	2010–2013	Taiwan	Two-phase treatment: 1–360 mg/kg for 2 h; 2–45 mg/kg for 0.5 h	~30–40 μg/g	NCT01173172	[66,67]
Melanoma	BPA	1987–2001	Japan	170–200 mg/kg for 3–5 h	7.3 μg/g	NA	[53]
Head and neck	Borofalan	2016–2019	Japan	Two-phase treatment: 1–200 mg/kg/h for 2 h; 2–100 mg/kg/h	32.8 ppm	jRCT2080224571	[54]

NA: not available.

**Table 2 cells-13-02065-t002:** In vivo studies of new boron delivery compounds (2022–2024).

Boron Carrier	Tumour Model	Reference
**BSH/BPA derivatives**		
Polymer–drug conjugate with BSH-conjugated side chains	Colon cancer	[90]
Fluorinated BPA derivatives	Melanoma	[91]
Fluorinated and α-methylated 3-borono-L-phenylalanine (3BPA) derivatives	Pancreatic adenocarcinoma	[92]
3-borono-L-tyrosine (BTS)	Head and neck cancer	[93]
**Boronated compounds**		
Boronated derivatives containing a carborane cage, a sulfamido group and an ureido functionality (CA-USF)	Mesothelioma	[94]
Boron–angiopep-2 peptide conjugates (notably ANG-B)	Glioma	[95]
**Targeted delivery systems**		
Multifunctional nanoliposome delivery system DOX-CB@lipo-pDNA-iRGD	Glioma	[96]
DPA-BSTPG targeting translocator protein (TSPO)	Glioma	[97]
Maleimide-functionalised closo-dodecaborate albumin conjugate (MID-AC)	Glioma	[98]
Cyclic arginine–glycine–aspartate (cRGD) targeting integrin αvβ3 added to MID-AC	Glioma	[99]
Anti-HER-2-antibody-conjugated carborane-integrated liposomes	Ovarian cancer	[100]
HER-2-targeted antibody-conjugated boron nitride nanotube/β-1,3-glucan complex	Ovarian cancer	[101]
*o*-carborane coupled to fibroblast activating protein (FAP) inhibitor (Carborane-FAPI)	Melanoma	[102]
Pteroyl-*closo*-dodecaborate-conjugated 4-(p-iodophenyl) butyric acid (PBC-IP)	Glioma	[103]
**Nanoparticles**		
^10^B-enriched boron phosphate nanoparticles (^10^BPO_4_ NPs)	Head and neck cancer	[104]
^10^B-enriched boron carbide (^10^B_4_C) nanoparticle functionalised with polyglycerol (PG) (^10^B_4_C-PG)	Colorectal carcinoma	[105]
Polyethylene-glycol-coated boron carbon oxynitride nanoparticles (PEG@BCNO)	Breast cancer	[106]
^10^B-enriched anti-EGFR-Gd^10^B_6_ nanoparticles	Head and neck cancer	[107]
Mesoporous silica nanoparticle (MSN)-based boron-containing agent coated with a lipid bilayer and decorated with SP94 peptide (SP94-LB@BA-MSN)	Liver cancer	[108]
^10^B-enriched hexagonal boron nitride nanoparticles grafted with poly(glycerol) (h-^10^BN-PG)	Colon carcinoma	[109]
Elemental boron (B) nanoparticles (BNPs)	Glioma	[110]
Gold nanoparticles (AuNPs) functionalised with cyclic arginine–glycine–aspartic acid (cRGD) peptides carrying BSH (AuNPs-BSH&PEGcRGD)	Glioma	[111]
Hydrophobic boron cluster carborane (CB)-integrated extracellular vesicles (CB@EVs)	Colon adenocarcinoma	[112]
**Small molecules**		
Boron rich block copolymer micelles	Melanoma	[113]
Boron-rich poly(ethylene glycol)-block-(poly(4-vinylphenyl boronate ester)) polymer micelles	Melanoma	[114]
Sodium salt of cobaltabis(dicarbollide) (Na[3,3′-Co(C_2_B_9_H_11_)_2_], abbreviated as Na[o-COSAN])	Oral cancer	[115]
Boron-conjugated 4-iodophenylbutanamide (BC-IP)	Glioma	[116]
^10^B-boronated derivative of temozolomide (TMZB)	Glioblastoma	[117]
**Dual purpose**		
4-mercaptophenylboronic acid (MPBA) linked to IR-780 dye	Liver cancer	[118]
Radioactive boron compounds ([^67^Ga]16 and [^125^I]17) conjugated to a closo-dodecaborate moiety ([B_12_H_12_]^2−^) with a gallium–DORA-c(RGDfK) complex, containing an RGD peptide	Glioblastoma	[119]
**Other compounds**		
Boronsome: carboranyl-phosphatidycholine-based liposome loaded with doxorubicin and olaparib	Breast cancer	[120]
Maleimide-functionalised *closo*-dodecaborate albumin conjugates (MID:BSA)	Oral Cancer	[121]
Carborane-based covalent organic framework (B-COF) loaded with immune adjuvant loaded with imiquimod	Melanoma	[122]
Gadolinium−boron-conjugated albumin (Gd-MID-BSA)	Colon carcinoma	[123]
Boron-containing carbon dot (BCD)–human serum albumin (HSA) complexes (BCD-HSA)	MelanomaProstate cancer	[124]

## Data Availability

No new data were created or analyzed in this study.

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
