# Peer review of "Current Insights into the Radiobiology of Boron Neutron Capture Therapy and the Potential for Further Improving Biological Effectiveness"

_cells, 2024, doi:10.3390/cells13242065_

Round 1
Reviewer 1 Report
Comments and Suggestions for Authors
This is a very comprehensive paper on BNCT based therapeutics.
The review is quite systematic with appropriate subtitles and is easy to grasp.
Authors need to add a translational aspect to their paper. A table with compounds in different stages of clinical evaluation e.g. L-boronophenylalanine (L-BPA) and sodium borocaptate (BSH) which are currently being used to treat patients is there. While there are compounds at different stages of clinical/animal trials, e.g. JHN002/borofalan. Including upcoming compounds is useful to understand upcoming trends of this newer technology
In conclusion sections, authors need to be judicious and clearly state the shortcomings of current BNCT, such as cancer recurrence rates.
Authors need to critically review that the abbreviations have their full forms cited when used for the first time, e.g. BSH
Pentagamaboronon-0 (PGB-0) is an important upcoming agent for the treatment of triple negative breast cancer. Authors need to add a section how BNCT can overcome treatment resistant cancers.
There is a new strategy to have dual acting BNCT agents that not only accumulate but also block an important cancer target. This greatly increases the efficacy of BNCT e.g. G protein-coupled receptor agonists and EGFR and kinase inhibitors.
Another trending direction is that the tight binding to enzymes overexpressed on the surface of tumor cells is sufficient for BNCT efficacy. This has been demonstrated for G-protein and MMP enzymes, which are specific targets of certain cancers.
All in all this is a very systematic and balanced review. It adds updated information lacking in current available reviewes. Authors need to improve translational and innovation aspects
Reviewer 2 Report
Comments and Suggestions for Authors
The title “Current insights into the radiobiology of boron neutron capture therapy and the potential for further improving biological 3 effectiveness.” is appropriate for this manuscript.
The manuscript discusses the current knowledge of the radiobiology of boron neutron capture therapy (BNCT) from in vitro and in vivo studies, particularly in the context of DNA damage and repair, and also presents evidence of established and novel boron-containing compounds aimed at improving the specificity and efficacy of treatment. However, the future perspectives, discussion and conclusions could be improved. In particular, it would be useful if the authors added some discussion parts on comparisons with other difficulties and the feasibility of radiobiology studies also in other fields such as Targeted Radionuclide Therapy with radiopharmaceuticals and radiolabeled nanoparticles.
Please see the comments below.
The manuscript addresses a hot topic by showing several radiobiology studies based on BNCT, the principle of which is to deliver a boron-containing drug that binds selectively to tumor cells and has a large cross-section capable of capturing a low-energy neutron. After administration of the boron-containing compound, the patient is exposed to a thermal or epithermal neutron beam. The compound enters an excited state after neutron capture and undergoes a nuclear fusion reaction to produce densely ionizing alpha particles. Since downstream, these particles release a quantity of energy in the tumor target, sparing healthy tissues, a comparison of this approach with a very similar one is required, which is represented by Targeted Radionuclide Therapy (TRT) with targeted radiopharmaceuticals.
1)The authors are strongly advised to consider expanding the discussion by following [10.3390/diagnostics13071210]. In this suggested manuscript, in fact, the authors discuss the importance and how to increase radiobiology and radiotheranostics studies through the use of increasingly sophisticated tools such as gamma counters, but also increasingly high-performance in vitro and in vivo tests, taking into account crossfire and bystander effects that sometimes do not allow quantitative estimates directly related to the samples. Additionally, artificial intelligence is discussed as an essential part of radiobiology workflows today. The authors are strongly advised to consider a comparison of BNCT and TRT. This will enrich the impact that this work will have in all branches of radiobiology and not only based on unsealed sources of ionizing radiation, but also sealed ones such as radionuclides.
2)In [10.1002/mp.16998] a novel radiation biology model based on nanodosimetry was proposed to accurately assess relative biological effectiveness and compound biological effectiveness for BNCT as it is currently not possible to accurately provide the biological effects of this therapy. It was revealed that both homogeneous and heterogeneous nanodosimetric parameters, as well as the corresponding biological model coefficients α and β, along with RBE values, show variations in response to different intracellular concentrations of Boron-10 isotope. In particular, the nanodosimetric parameter effectively captures the fluctuations in the model coefficients α and RBE. To enhance the impact of the study in contemporary research, it is suggested that the authors also consider this very recent experimental evidence in their discussions.
3)To make reading the document easier for the reader, an Abbreviations section should be added to the end of the document.
4) Finally, to make the article more coherent, the references should be extended, and English should be fluent throughout.
Reviewer 3 Report
Comments and Suggestions for Authors
Comments
Journal: Cells
Article type: Review
Article ID: Cells-3310537
This work is related to the applicability and challenges of BNCT as an alternative to X-ray radiotherapy. The review is well-written and the sections well described. The importance and ongoing studies of BNCT are highlighted and the figures and tables important. The reviewer suggests that an additional table summarizing the new delivery agents would help to better understand what is being investigated in a very sensitive field that is the specific delivery of compounds.
Minor comments:
Line 36 – What is the difference between using drugs or conjugates?
Line 40 – Rephrase because it is not clear
Line 91 – CDD already described in line 51
Line 129 – indent the paragraph
Line 286 – The clinical trials code should be showed in the table.
Line 454 – A table summarizing the new delivery agents is a plus, since it is an important part of the review
Line 664 – Although this is a review on the applicability of BNCT, a section or paragraph highlighting the limitations and future perspectives of this approach is important
Major comment:
For a review, it lacks references of the last 5 years.
Round 2
Reviewer 1 Report
Comments and Suggestions for Authors
All the comments from first round of review have been addressed
Author Response
We thank the Reviewer for commenting that all points have been addressed.
Reviewer 2 Report
Comments and Suggestions for Authors
The Authors partially respond to the comments expressed.
they simply insert two references for comment 1 without having read the suggested work at all. They insert a very dated reference dating back to 2020 [10.1053/j.semnuclmed.2019.07.006] on a topic that is strongly evolving from year to year, without considering or reading the proposed manuscript [10.3390/diagnostics13071210] much more recent and precisely from 2023 with very current implications and on radionuclides of global interest (64Cu-, 68Ga-, 125I-, and 99mTc).
Please replace reference 146 [10.1053/j.semnuclmed.2019.07.006] with reference [10.3390/diagnostics13071210].
For comment 2 they reply that it is far from the scope of the manuscript, even if it is true they should have at least mentioned something in the discussions perhaps putting it within the limits of the manuscript.
Round 3
Reviewer 2 Report
Comments and Suggestions for Authors
The Authors answered all the questions.
The paper is worthy of publication